# Carbonic Anhydrase IX-Targeted α-Radionuclide Therapy with 225Ac Inhibits Tumor Growth in a Renal Cell Carcinoma Model

**DOI:** 10.3390/ph15050570

**Published:** 2022-05-02

**Authors:** Robin I. J. Merkx, Mark Rijpkema, Gerben M. Franssen, Annemarie Kip, Bart Smeets, Alfred Morgenstern, Frank Bruchertseifer, Eddie Yan, Michael P. Wheatcroft, Egbert Oosterwijk, Peter F. A. Mulders, Sandra Heskamp

**Affiliations:** 1Department of Medical Imaging, Nuclear Medicine, Radboudumc, 6525 Nijmegen, The Netherlands; mark.rijpkema@radboudumc.nl (M.R.); gerben.franssen@radboudumc.nl (G.M.F.); annemarie.kip@radboudumc.nl (A.K.); sandra.heskamp@radboudumc.nl (S.H.); 2Department of Urology, Radboudumc, 6525 Nijmegen, The Netherlands; egbert.oosterwijk@radboudumc.nl (E.O.); peter.mulders@radboudumc.nl (P.F.A.M.); 3Department of Pathology, Radboudumc, 6525 Nijmegen, The Netherlands; bart.smeets@radboudumc.nl; 4European Commission, Joint Research Centre (JRC), 76344 Karlsruhe, Germany; alfred.morgenstern@ec.europa.eu (A.M.); frank.bruchertseifer@ec.europa.eu (F.B.); 5Telix Pharmaceuticals Limited, Melbourne, VIC 3051, Australia; eddie.yan@telixpharma.com (E.Y.); mike@telixpharma.com (M.P.W.)

**Keywords:** radioimmunotherapy, RCC, actinium-225, CAIX, G250

## Abstract

In this study, we compared the tumor-targeting properties, therapeutic efficacy, and tolerability of the humanized anti-CAIX antibody (hG250) labeled with either the α-emitter actinium-225 (^225^Ac) or the β^-^-emitter lutetium-177 (^177^Lu) in mice. BALB/c nude mice were grafted with human renal cell carcinoma SK-RC-52 cells and intravenously injected with 30 µg [^225^Ac] Ac-DOTA-hG250 (^225^Ac-hG250) or 30 µg [^177^Lu] Lu-DOTA-hG250 (^177^Lu-hG250), followed by ex vivo biodistribution studies. Therapeutic efficacy was evaluated in mice receiving 5, 15, and 25 kBq of ^225^Ac-hG250; 13 MBq of ^177^Lu-hG250; or no treatment. Tolerability was evaluated in non-tumor-bearing animals. High tumor uptake of both radioimmunoconjugates was observed and increased up to day 7 (212.8 ± 50.2 %IA/g vs. 101.0 ± 18.4 %IA/g for ^225^Ac-hG250 and ^177^Lu-hG250, respectively). Survival was significantly prolonged in mice treated with 15 kBq ^225^Ac-hG250, 25 kBq ^225^Ac-hG250, and 13 MBq ^177^Lu-hG250 compared to untreated control (*p* < 0.05). Non-tumor-bearing mice that received single-dose treatment with 15 or 25 kBq ^225^Ac-hG250 showed weight loss at the end of the experiment (day 126), and immunohistochemical analysis suggested radiation-induced nephrotoxicity. These results demonstrate the therapeutic potential of CAIX-targeted α-therapy in renal cell carcinoma. Future studies are required to find an optimal balance between therapeutic efficacy and toxicity.

## 1. Introduction

Renal cell carcinoma (RCC) accounts for 5% and 3% of all cancers worldwide for men and women, respectively [1]. RCC comprises a heterogenous group of malignancies of which clear cell RCC (ccRCC) is most common. With the paradigm of cancer therapy shifting towards personalized medicine, the demand for tumor-specific targeting approaches is increasing. In ccRCC, the tumor-associated antigen carbonic anhydrase IX (CAIX) is expressed ubiquitously in both primary tumor and metastases with very limited expression in normal tissue [2]. The monoclonal antibody G250 specifically targets CAIX and is internalized after binding [3]. Hence, G250 has been extensively investigated both for radioimmunoimaging and radioimmunotherapy (RIT) in ccRCC [4,5,6]. Although ccRCC is historically seen as a tumor with an intrinsic high radio-resistance, studies with high-dose ablative radiotherapy reject this notion [7]. This renewed insight is in accordance with previous RIT studies, using [^177^Lu]Lu-DOTA-cG250, that showed a promising initial clinical response in patients with progressive metastatic ccRCC. Unfortunately, bone marrow depletion and consequently myelotoxicity has thus far hampered the clinical implementation of this treatment [8].

A promising alternative to RIT with β^−^ -emitters is targeted α-therapy using radionuclides such as actinium-225 (^225^Ac; t_1/2_ = 9.9 days), lead-212 (^212^Pb t_1/2_ = 10.6 h), or bismuth-213 (^213^Bi; t_1/2_ = 45.6 min), since α-particles have a significantly higher linear energy transfer (LET) over a shorter distance compared to β^−^ -emitters [9].

While multiple α-emitters have become available over the last decade, antibody-targeted therapies require significant time to accumulate in the tumor (3–7 days), and thus a long-lived radionuclide such as ^225^Ac is considered a good fit. ^225^Ac decays via the α-emitting daughter nuclides francium-221 (t_1/2_ = 5 min), astatine-217 (t_1/2_ = 32.3 ms), and ^213^Bi, to the stable 2^09^Bi. Herein lies its greatest therapeutic potential, since rapid internalization of the radioimmunoconjugate by the cancer cells, followed by intracellular decay of four α-particles, could effectuate a very high tumor radiation dose [10]. However, during the decay of ^225^Ac, its α-emitting daughter is decoupled from the targeting agent due to the recoil energy exceeding the energy of the chelating bond. Subsequently, daughter radionuclides can redistribute throughout the body leading to off-target toxicity. The combination of long circulating antibodies with uncontrolled α-decay by daughter nuclides can pose a significant risk. Therefore, it is paramount to find the therapeutic window for targeted α-therapy that results in minimal toxicity [11].

To our knowledge, so far, no studies have been performed using CAIX-targeted α-therapy in renal cell carcinoma. The aim of our study was to evaluate the tumor-targeting properties, therapeutic efficacy, and tolerability of [^225^Ac]Ac-DOTA-hG250 (^225^Ac-hG250) RIT in comparison to [^177^Lu]Lu-DOTA-hG250 (^177^Lu-hG250) in mice bearing CAIX-expressing human ccRCC xenografts.

## 2. Results

### 2.1. Biodistribution

Tumor uptake of ^225^Ac-hG250 and ^177^Lu-hG250 was high and increased over time until at least 7 days p.i. (Figure 1). Tumor accumulation was significantly higher in the ^225^Ac-hG250 group compared to the ^177^Lu-hG250 group (*day 3*: 110.6 ± 34.6 %IA/g vs. 66.4 ± 18.3 %IA/g, (*p* < 0.05), *day 7*: 212.8 ± 50.2 %IA/g vs. 100.5 ± 18.3 %IA/g (*p* < 0.05), respectively). The blood level of both radioimmunoconjugates decreased over time and was significantly higher for ^225^Ac-hG250 compared with ^177^Lu-hG250 at day 1 and day 3 p.i. (*day 1*: 18.7 ± 0.7 %IA/g vs. 14.0 ± 2.7 %IA/g (*p* < 0.001), *day 3:* 13.9 ± 2.1 %IA/g vs. 10.9 ± 1.2 %IA/g (*p* = 0.025), respectively). There was no significant difference in blood levels at day 7 (4.7 ± 2.7 %IA/g vs. 4.1 ± 1.4 %IA/g; *p* = 0.66). Similarly, in well-vascularized organs such as the lung, kidney, and heart, the uptake of ^225^Ac-hG250 was higher at early timepoints compared with ^177^Lu-hG250 and normalized at day 7.

During dissection, the macroscopic evaluation of the spleens in mice that received 50 kBq ^225^Ac-hG250 showed paleness of the organ and clear signs of declining organ mass, potentially driven by atrophy of the organ (Appendix A).

A high relative uptake of ^213^Bi was observed in the kidneys (52.1 ± 14.6 %IA/g, 68.3 ± 6.7 %IA/g, and 17.3 ± 4.9 %IA/g for days 1, 3, and 7, respectively) at time of dissection (Figure 2A). In contrast, only a minimal amount of ^213^Bi was present in the circulation at time of dissection (Figure 2B). In both the liver and tumor, uptake of ^213^Bi was marginally lower at time of dissection, compared to equilibrium state (Figure 2C,D).

### 2.2. Therapy Study

Radionuclide therapy with hG250 reduced tumor growth and prolonged survival in tumor-bearing mice. Mean tumor doubling time in mice that received 5 kBq ^225^Ac-hG250 was 67.8 ± 15.7 days compared with 31.9 ± 13.4 days for untreated mice (*p* < 0.001) (Figure 3). Of note, in two mice that received 5 kBq ^225^Ac-hG250, tumor size decreased over time, and therefore they were not included in the analysis of tumor doubling time. Similarly, mean tumor doubling time could not be calculated for the 15 kBq ^225^Ac-hG250, 25 kBq ^225^Ac-hG250, and 13 MBq ^177^Lu-hG250 groups due to an overall decrease in tumor size. The mean tumor size of mice that received either 15 or 25 kBq of ^225^Ac-hG250 showed no significant difference compared to the 13 MBq ^177^Lu-hG250 group at end of study (150 days post-inoculation) (62.3 ± 79.9, 44.1 ± 19.7, and 44.0 ± 50.6, respectively; *p* = 0.745). Survival was significantly prolonged in mice treated with 15 kBq ^225^Ac-hG250, 25 kBq ^225^Ac-hG250, and 13 MBq ^177^Lu-hG250 (median survival time >150 days) compared to control (median survival time = 99 days, *p* < 0.05), but not in mice treated with 5 kBq ^225^Ac-hG250 (median survival time = 149 days, *p* = 0.47).

Immunohistochemical analysis of tumor sections indicated a dose-dependent radiation effect after radionuclide-hG250 treatment at 150 days post-inoculation. The H&E-staining tumor specimens of mice treated with 15, 25 kBq ^225^Ac-hG250 or 13 MBq ^177^Lu-hG250 displayed late radiation effects, exhibited by enlarged nuclei, multinucleation, and intercellular fibrosis (Figure 4). This effect was not prominent in mice that received 5 kBq ^225^Ac-hG250. Additionally, we observed that membrane expression of CAIX remained present across all treatment groups.

### 2.3. Tolerability Study

^225^Ac-hG250 treatment showed a dose-dependent effect on body weight in non-tumor-bearing mice (Appendix A). In the 15 and 25 kBq ^225^Ac-hG250 groups, 4/6 and 3/6 non-tumor-bearing mice were taken out of the experiment due to severe weight loss and clinical deterioration as judged by a blinded biotechnician (loss in body weight: 18.6 ± 3.4% and 14.2 ± 2.1%, respectively, at week 17). In the remaining groups, no significant changes in weight and clinical status were observed.

Hemocytometry and serum chemistry showed no significant differences in hemoglobin, leucocyte, ALAT, ASAT, or creatinine in comparison to baseline measurement across all treatment groups (Figure 5). Of note, platelet levels were elevated 12 weeks p.i. in all groups. Since this occurred similarly across all groups, it was not considered to be treatment related.

Mice that received treatment with 15 or 25 kBq ^225^Ac-hG250 showed an impaired uptake of radiolabeled DMSA after 10 weeks on SPECT imaging (Figure 6C,D). Quantification of the radioactive uptake of [^99m^Tc] Tc-DMSA (^99m^Tc-DMSA) in the kidneys demonstrated a significant decrease at 16 weeks p.i. for mice that received 15 or 25 kBq ^225^Ac-hG250, compared with the non-treated control group (14.5 %IA/g, 17.0 ± 2.8 %IA/g, and 61.1 ± 3.7 %IA/g, respectively). Immunohistochemical analyses of the kidneys showed tubular abnormalities in mice treated with 15 kBq or 25 kBq ^225^Ac-hG250, evidenced by tubular necrosis, denudation of tubular basement membrane, interstitial fibrosis, and tubular atrophy (IFTA) (Figure 6A and Appendix A). Tubular necrosis in mice treated with 15 or 25 kBq ^225^Ac-hG250 varied from grade 4 to 5, which was significantly higher compared with mice treated with 13 MBq ^177^Lu-hG250 (grade 0–1) or control (grade 0) (Figure 6B). IFTA was scored mild in mice treated with 15 or 25 kBq ^225^Ac-hG250 (grades 1–4) and was not observed in the remaining groups.

## 3. Discussion

In the present study, we show that an ^225^Ac-labeled CAIX-targeted antibody can effectively target and treat ccRCC xenografts in mice. However, we also observed dose-dependent radiation nephropathy.

Our data demonstrate a very high relative tumor uptake of ^225^Ac-hG250, which was significantly higher than that of ^177^Lu-hG250. A possible explanation could be that targeted α-therapy induces an early onset alteration in the tumor environment, possibly augmenting the enhanced permeability and retention (EPR) of antibodies in tumor tissue [12]. Although previous studies have demonstrated that the uptake of G250 in SK-RC-52 tumors is predominantly CAIX-mediated, an ^225^Ac-labeled irrelevant control antibody could aid in distinguishing between CAIX-mediated and a-specific tumor uptake in this setting [13]. The potential changes in the tumor environment may have been caused by the fact that we injected a relative high activity dose of ^225^Ac-hG250. However, this was required to accurately measure γ-decay of ^225^Ac-derived daughter radionuclides in a γ-counter at later time points (i.e., day 7). Mice in our biodistribution study received 50 kBq ^225^Ac-hG250, which was 2- to 10-fold higher than the dose administered in the therapy study.

Furthermore, spleen atrophy was observed in mice that received 50 kBq ^225^Ac-hG250. Even though there is no CAIX expression in the spleen, physiological uptake of radiolabeled antibodies due to the fenestrated vasculature in the spleen has been described previously [14]. Since the hematopoietic system is highly radiosensitive and α-emitters have high relative biological effectiveness (RBE), a minimal exposure of α-emission to the spleen could induce radiotoxicity [15]. However, importantly, no signs of hematopoietic toxicity were observed in mice receiving 5, 15, or 25 kBq ^225^Ac-hG250 for up to 150 days. Other small differences in the biodistribution of ^225^Ac-hG250 and ^177^Lu-hG250 were observed, for example, in blood and liver. To explore the underlying mechanism of these differences, more research is needed.

One of the main challenges in targeted α-therapy using ^225^Ac is the uncertainty about the fate of the unchelated daughter radionuclides in vivo. In our study, we aimed to determine the localization and potential redistribution of ^213^Bi in different tissues. We analyzed this by calculating the amount of ^213^Bi in tissue at time of dissection and compared this with a late measurement of ^213^Bi (equilibrium state), which contains the amount of ^213^Bi activity originating from ^225^Ac present in the sample [16]. We observed that the relative uptake of ^213^Bi in the liver and tumor was not significantly different at time of dissection compared to equilibrium state. This suggests that there is limited relocation of ^213^Bi from the liver and tumor cells to other tissues after decay of ^225^Ac. Bismuth is known to accumulate in the renal cortex, specifically in the proximal tubular cells [17,18]. This was confirmed by our study, as we observed that uptake of ^213^Bi in the kidneys was much higher at time of dissection, compared with measurements at equilibrium state. These results indicate that a large fraction of ^213^Bi in the kidneys does not originate from ^225^Ac in the kidneys, but also from ^225^Ac decaying elsewhere. This could be for example from blood, as there the opposite was observed. At time of dissection, the concentration of ^213^Bi was much lower than expected according to the equilibrium measurements.

Previous studies in both primates and mice suggested that radiation nephropathy can occur after treatment with ^225^Ac-labeled antibodies [19,20]. Schwartz et al. showed that the α-particles emitted by ^213^Bi are responsible for the largest fraction of the absorbed kidney dose, specifically in the medulla, after ^225^Ac-mAb administration [16]. In order to reduce radiation induced nephropathy, multiple nephroprotective strategies have been studied. Jaggi et al. evaluated the effect of different diuretic agents and metal chelating products on the development of radiation nephropathy after injection of ^225^Ac and concluded that inhibition of the renin–angiotensin–aldosterone system could be a viable method to provide renal protection in targeted α-therapy [20].

To evaluate whether treatment with ^225^Ac-hG250 leads to late onset radiation nephropathy, we performed ^99m^Tc-DMSA imaging. The renal uptake of ^99m^Tc-DMSA protein complex is a megalin/cubulin-mediated endocytosis process that occurs in the proximal tubulus [21]. Similarly, bismuth is delivered to the kidney bound to a metallothionein (MT) followed by a megalin-mediated renal uptake of the bismuth–MT complex [22]. Hence, imaging of the proximal tubulus function could serve as a marker for bismuth-induced nephropathy. The renal impairment demonstrated by ^99m^Tc-DMSA imaging was in concordance with the histopathological evaluation in our study. This confirms that ^99m^Tc-DMSA imaging might provide an elegant non-invasive method to quantify and monitor nephropathy in preclinical models. Furthermore, this method has been shown to be more sensitive than serum creatinine levels in evaluating radiation nephropathy [23].

Histopathological evaluation of the kidneys of mice that received 5 kBq ^225^Ac-hG250 revealed no sign of renal nephropathy, indicating that the radiation nephropathy is dose-dependent. Even though treatment with 5 kBq ^225^Ac-hG250 did not lead to a survival benefit, a tumor growth-inhibiting effect that lasted for approximately 50 days was seen. Additionally, the histopathological evaluation of treated tumors showed that transmembranous CAIX expression remained present. These findings provide rationale for fractionated dosing patterns.

A recent study by Minnix et al. confirms this by demonstrating that a multi-treatment regimen that consisted of a single 1.85 kBq dose followed by five doses of 0.70 kBq [^225^Ac] Ac-DOTA-huCC49 (^225^Ac-huCC49) lead to similar survival as a single 7.4 kBq dose of ^225^Ac-huCC49 in an ovarian cancer mice model (86.0 and 104.0 days, respectively) [24]. Moreover, the multi-treatment regimen of low dose ^225^Ac-huCC49 showed an alleviating effect on whole body toxicity. Although a direct comparison between dosing in different mice models is difficult, this suggests that alternative dosing regimens could result in less toxicity while remaining effective.

Most importantly, we demonstrate that mice treated with 15 and 25 kBq of ^225^Ac-hG250 show significant tumor reduction up to 18 weeks post-treatment, leading to an extended survival. While the tumor growth inhibition of high doses of ^225^Ac-hG250 treatment was on par with ^177^Lu-hG250 treatment, the morphological and immunohistochemical profiles of the treated tumors suggest that the radiation damage is more severe in tumors treated with 15 kBq an 25 kBq ^225^Ac-hG250 compared with 13 MBq ^177^Lu-hG250.

Our study confirms the therapeutic potential of targeted α-therapy by demonstrating that mice bearing SK-RC-52 tumors can be treated effectively with single doses of ^225^Ac-hG250. Studies that aim to investigate whether the same therapeutic effect can be established in other models with different parameters (i.e., different CAIX receptor expression levels or different radiosensitivity) have yet to be conducted but could provide valuable information in pursuit of clinical implementation.

Lastly, it is uncertain as to what degree the radiation-induced nephropathy in mice can be accurately translated to a clinical setting. In a phase I study by Jurcic et al., patients with relapsed or refractory AML were successfully treated with 18.5–148 kBq/kg ^225^Ac-lintuzumab, without signs of radiation-induced nephrotoxicity. In the following phase I/II study, patients receiving ^225^Ac-lintuzumab were co-administered with furosemide and spironolactone to prevent radiation-induced nephrotoxicity, and thus far, no signs thereof have been reported [25]. These data suggest that nephrotoxicity of antibody-based targeted α-therapy may be less of an issue when preventive measures that focus on improving diuresis are implemented, but thorough research on this matter is paramount.

In conclusion, targeted α-therapy is an emerging class of anti-cancer treatment that, in the rapidly evolving landscape of RIT, could serve either as an alternative or as an addition to β^−^ -emitting RIT. This study supports the perspective that targeted α-therapy could play a role in the future of ccRCC treatment. However, before clinical implementation of ^225^Ac-based treatment in RCC can be considered, it is essential to reduce the accumulation of daughter nuclide in organs at risk. Here, multiple strategies that can be applied such as the previously discussed multi-dose regimens and nephroprotective drugs. Alternatively, use of agents that are more rapidly cleared (i.e., small molecules, peptides, or antibody-F(ab’)_2_ fragments) could potentially lead to lower toxicity. However, these agents are often accompanied by a lower tumor uptake compared to their target specific counterpart. As a result, a higher activity may be required to achieve a similar tumor absorbed dose, possibly offsetting the benefits. Therefore, a careful balance is needed between therapeutic efficacy and toxicity. Clinical trials involving [^225^Ac] Ac-PSMA-617 have demonstrated the feasibility of combining multi-treatment regimens with use of a rapid-clearing PSMA ligands [26].

## 4. Material and Methods

### 4.1. Antibodies and Cell Lines

Humanized G250 (hG250) is an IgG1 monoclonal antibody that is directed against the CAIX antigen and was received as a kind gift from Telix Pharmaceuticals Ltd. (Melbourne, VIC, Australia).

The CAIX-expressing RCC cell line SK-RC-52 (obtained from Memorial Sloan Kettering Cancer Center, New York, NY, US; RRID: CVCL_6198) was derived from a mediastinal metastasis of a primary RCC [27]. SK-RC-52 cells were cultured in RPMI-1640 medium, supplemented with 10% fetal calf serum (FCS) at 37 °C in a humidified atmosphere with 5% CO_2_. Prior to in vitro or in vivo experiments, cells were washed with saline, trypsinized, and washed with RPMI-1640 10% FCS.

### 4.2. Conjugation, Radiolabeling, and Quality Control

HG250 was conjugated with S-2-(4-isothiocyanatobenzyl)-1,4,7,10-tetraazacyclododecane tetraacetic acid (p-SCN-Bn-DOTA, Macrocyclics™, Plano, TX, USA) as described previously [28]. A total of 87.7 MBq ^225^Ac (received as a kind gift from Joint Research Centre, Karlsruhe, Germany) was dissolved in 390 µL of 0.01 M HCl to obtain a concentration of 0.22 MBq/µL. No-carrier-added ^177^Lu (concentration 16 MBq/µL) was obtained from ITM Medical Isotopes GmbH (Garching, Germany).

DOTA-conjugated hG250 was incubated with ^225^Ac in 0.1 M TRIS buffer, pH 9.0, at 37 °C, under strict metal-free conditions for 60 min. Radiochemical purity was determined directly after incubation by instant thin layer chromatography (ITLC), using ITLC silica gel strips (Agilent Technologies, Santa Clara, CA, USA), and 0.1 M of citrate buffer (pH 6.0) for [^225^Ac] Ac-DOTA-hG250. Labeling efficiency was 99.3% at maximum specific activity of 0.0157 MBq/µg. The determination of the immunoreactive fraction of ^225^Ac-hG250 in relation to SK-RC-52 cells was performed as described by Lindmo et al. and exceeded 80% [29].

DOTA-conjugated hG250 was incubated with ^177^Lu in 0.5 M MES buffer, pH 5.5, at 37 °C, under strict metal-free conditions for 30 min. After incubation, 50 mM ethylenediaminetetraacetic acid (EDTA) was added to a final concentration of 5 mM to complex nonincorporated ^177^Lu. The radiochemical purity of [^177^Lu] Lu-DOTA-hG250 was determined as described above, and labeling efficiency was 99.0% at maximum specific activity of 0.79 MBq/µg. The immunoreactive fraction of ^177^Lu-hG250 exceeded 80%.

### 4.3. Animal Experiments

The experiments were performed in female BALB/cAnNRj-Foxn1^nu/nu^ mice (6–8 weeks old) with a median weight of 20.4 g (range 17.6–23.0) (Janvier, le Genest-Saint-Isle, France). Mice were housed in individualized ventilated cages with ad libitum access to animal chow and water. Mice were engrafted subcutaneously with 3 × 10^6^ SK-RC-52 cells in 0.2 mL of RPMI-1640 in the right flank. Three weeks after cell inoculation, tumors reached a size of 30–200 mm^3^.

### 4.4. Biodistribution Studies

To compare the biodistribution of ^177^Lu-hG250 and ^225^Ac-hG250, SK-RC-52 tumor-bearing mice with a median tumor volume of 75.1 mm^3^ (range 33.1–180.7) were randomized into groups of 5 mice through block randomization. Mice were injected intravenously with 0.2 MBq ^177^Lu-hG250 or 50 kBq ^225^Ac-hG250, each diluted in PBS containing 0.5% BSA. All mice received a protein dose of 30 µg hG250 (volume, 200 µL/mouse). Mice were sacrificed at 1, 3, or 7 days post-injection (p.i.). The tumor and normal tissues were harvested, weighed, and counted in a γ-counter (1480 Wizard 3; LKB/Wallace, Perkin Elmer, Boston, MA, USA). To correct for radioactive decay, three aliquots containing 1% of the injected dose were counted simultaneously. The activity in samples was expressed as percent injected activity per gram tissue (%IA/g) and reported as mean ± standard deviation (SD).

The decay of ^225^Ac generates only weak γ-emissions that cannot be quantified in a straightforward manner, and therefore the decay of its progeny (i.e., ^221^Fr: 218 keV, counting window 170–270 keV) was counted once the measurement reflected only activity that was originally present in the sample as ^225^Ac (after at least 24 h). This method has been previously described by Schwartz et al. and Kruijff et al. [16].

In order to estimate the in vivo distribution of ^213^Bi (440 keV, counting window 380–520 keV), tumor blood, kidney, and liver were transferred to the γ-counter directly after dissection and measured continuously for approximately 12 h. The ^213^Bi activity in specified organs at time of sacrifice (t_0_) was curve-fitted by using the non-linear one phase decay fit in GraphPad Prism version 5.03 (GraphPad Software, Inc., San Diego, CA, USA). Here, we again assume that all free ^213^Bi that was present at the moment of sacrifice has decayed at late measurements (beyond 10 half-lives) and all remaining ^213^Bi is generated by ^225^Ac originally present in the sample.

### 4.5. Therapy and Tolerability Studies

To study the therapeutic efficacy of ^225^Ac-hG250 and ^177^Lu-hG250, SK-RC-52-tumor-bearing mice with a median tumor volume of 82.9 mm^3^ (range 28.9–156.8 mm^3^) were randomly divided in groups of 10 mice through block randomization. Mice in the treatment groups were injected intravenously with 30 µg of either 5, 15, or 25 kBq of ^225^Ac-hG250 or 13 MBq ^177^Lu-hG250. The activity doses of ^225^Ac-hG250 were based on unpublished pilot data, while the 13 MBq of ^177^Lu-hG250 is based on a previous optimization study [13]. Mice in the control group received no treatment. Primary outcomes were tumor growth rate and survival. Tumor growth rate was expressed as doubling tumor time, calculated by fitting an exponential growth equation. During the entire experiment, the biotechnicians were blinded for the treatment.

The tumor size was measured twice a week with a caliper during the entire experiment. Tumor volume was calculated using an ellipsoid model with the following formula:$$V=\frac{4}{3}\pi abc$$
in which *a*, *b*, and *c* are the tumor radii. Mice were removed when reaching predefined humane endpoints, which included tumor volume ≥2 cm^3^, ulcerative tumor growth, weight loss >15% within 2 d, weight loss >20% compared with baseline, and severe clinical deterioration as assessed by a biotechnician. At the end of experiment, which was predefined at 150 days post-inoculation, all animals were inspected for macroscopic evidence of abnormalities, and the tumors were harvested, fixated in 4% formalin, embedded in paraffin, and stained with H&E and CAIX. Survival curves according to the Kaplan–Meier method were generated for each group.

In order to evaluate both short- and long-term toxicity, non-tumor-bearing mice with a median weight of 21.8 g (range 19.4–24.0 g) were divided randomly in groups of 6 mice. Similar to the therapy study, mice were injected intravenously with 30 µg of either 5, 15, or 25 kBq of ^225^Ac-hG250 or 13 MBq ^177^Lu-hG250 and followed for up to 18 weeks post-treatment. Again, a control group receiving no treatment was included. The mice were weighed weekly and sacrificed when reaching end of study or humane endpoints as determined by a biotechnician. For the evaluation of liver function, kidney function, and hematological status, blood sampling was performed at baseline, 6 weeks, and 12 weeks p.i. Mice in each group were further divided in groups of 3 mice (A & B). In group A, blood was analyzed on alanine-aminotransferase (ALAT), aspartate aminotransferase (ASAT), and creatinine. Additionally, these mice received [^99m^Tc] Tc-DMSA functional renal imaging at baseline, 10, and 16 weeks p.i. In group B, blood was analyzed on hemoglobin, leucocytes, and platelets. After 18 weeks, all mice were sacrificed, and the kidneys were harvested, fixated in 4% formalin, and embedded in paraffin before being stained with periodic acid-Schiff (PAS) for further analysis. All kidney sections were scored by a renal pathology expert (BS) on a 5-point scale in terms of tubular necrosis, interstitial fibrosis, and tubular atrophy (IFTA).

### 4.6. Renal Imaging

To monitor the renal function in vivo, mice in the tolerability study underwent SPECT imaging using ^99m^Tc-labeled dimercaptosuccinic acid (DMSA) at baseline, 10 weeks p.i., and 16 weeks p.i. Preparation of [^99m^Tc] Tc-DMSA (^99m^Tc-DMSA) was performed as per the manufacturer’s protocol (Curium Netherlands B.V., Petten, the Netherlands). In short, 740–1100 MBq of pertechnetate was added to 1.2 mg of DMSA and incubated for 15 min at 25 °C. Mice received an intravenous tail injection of 20 MBq ^99m^Tc-DMSA in 200 µL saline 2 h prior to SPECT imaging. The SPECT scans were acquired with the U-SPECT-II/CT (MILabs, Utrecht, the Netherlands) using a 1.0 mm diameter pinhole mouse high sensitivity collimator and acquisition time of 15 min. Image reconstruction was performed with MILabs reconstruction software using a 16-subset expectation maximization algorithm with a voxel size of 0.2 mm and 3 iterations. Quantification of renal uptake of ^99m^Tc-DMSA was performed by drawing volumes of interest (VOIs) around the kidneys. The radioactivity was corrected for decay and volume of the VOI, which resulted in a percent injected activity per gram kidney tissue, assuming a tissue density of 1.0 g/cm^3^.

### 4.7. Histology and Immunohistochemistry

Assessment of the kidneys. Paraffin-embedded tissue sections (4 µm) were stained with periodic acid-Schiff (PAS) staining and H&E. All kidneys were scored by a dedicated nephropathologist (BS) on a 5-point scale in terms of tubular necrosis, interstitial fibrosis, and tubular atrophy (IFTA).

Assessment of the tumors. All tumor tissues were stained for H&E and CAIX. For CAIX staining, paraffin-embedded tumor sections (4 µm) were deparaffinized. Antigen retrieval was performed by heating the tissue to 96 °C in a 10 mM citrate (pH 6.0) buffer. Next, endogenous peroxidase activity was blocked by incubating the tissue in 3% hydrogen peroxidase in 10 mM phosphate-buffered saline (PBS) for 10 min at room temperature. Subsequently, tissue sections were stained by using the commercially available Mouse-on-Mouse immunodetection kits (Vector Laboratories, Burlingame, CA, USA) with M75 (1:10,000 dilution), anti-CAIX, as the primary antibody. The CAIX staining was visualized with Bright-DAB (Immunologic, Duiven, The Netherlands) by incubating for 8 min at room temperature. All sections were counterstained with hematoxylin for 5 s, dehydrated in ethanol, and mounted. Tumor tissue was evaluated by a urological cancer pathologist on morphological and pathological deviations.

### 4.8. Statistical Analysis

For the analysis of comparing means between two groups, significance was tested using the independent t-test and were considered significant at *p* < 0.05. In the analysis of comparing means between more than two groups, ANOVA was used. Additionally, Levene’s test for homogeneity of variances was performed to verify the assumption of equal variances. Differences in renal toxicity were tested for significance using the nonparametric Kruskal–Wallis test and were considered significant at *p* < 0.05, two sided. Kaplan–Meier survival curves were analyzed for differences using the log-rank test and considered significant when *p* < 0.05, two sided. Analyses were performed using the software package SPSS version 25.0 (IBM, Armonk, NY, USA).

## 5. Conclusions

^225^Ac-hG250 showed high accumulation in subcutaneous CAIX-expressing SK-RC-52 xenografts. The biodistribution of ^225^Ac-hG250 and ^177^Lu-hG250 was similar with high tumor-to-normal tissue ratios and low uptake in non-target organs. Therapy studies with ^225^Ac-hG250 showed a dose-dependent response on tumor growth inhibition and survival rates. However, dose-dependent renal toxicity was observed in mice that received ^225^Ac-hG250. Future studies aimed at finding the optimal balance between efficacy and toxicity are required (i.e., fractionated dosing regimens).

## Figures and Tables

**Figure 1 pharmaceuticals-15-00570-f001:**
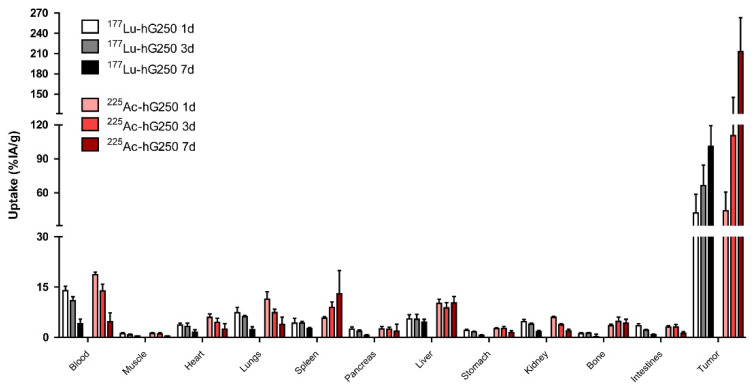
Biodistribution of radiolabeled hG250 in SK-RC-52 tumor-bearing mice at 1, 3, and 7 days after injection (*n* = 5).

**Figure 2 pharmaceuticals-15-00570-f002:**
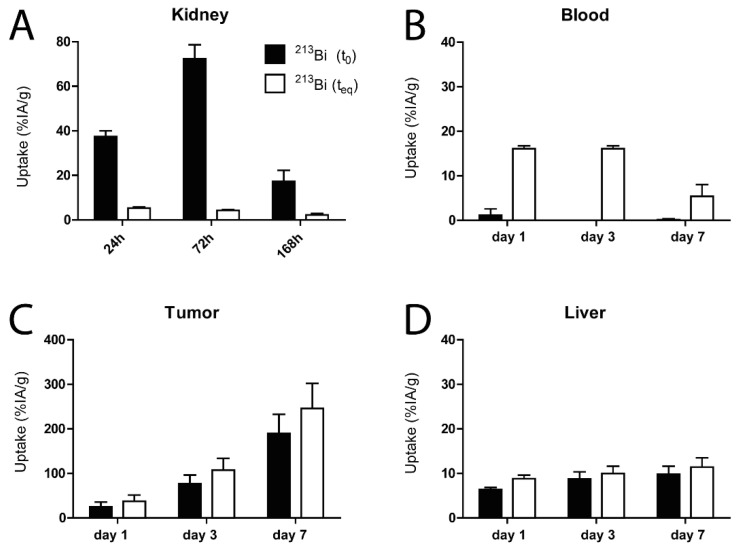
The calculated uptake of ^213^Bi at time of dissection (t_0_) in comparison to uptake of ^213^Bi at equilibrium state (t_eq_: ~24 h after dissection) in organs of interest in SK-RC-52 tumor-bearing mice at 24, 72, or 168 h p.i. (**A**) Uptake in the kidneys. (**B**) Uptake in the blood. (**C**) Uptake in the tumor. (**D**) Uptake in the liver. p.i.: post injection.

**Figure 3 pharmaceuticals-15-00570-f003:**
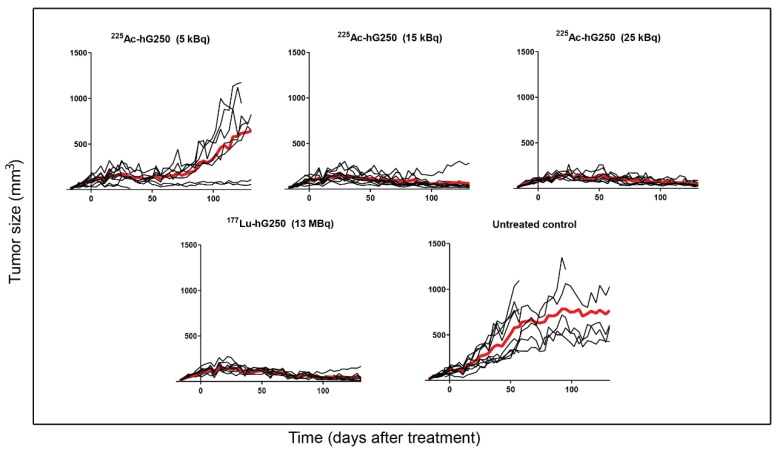
The tumor size over time in SK-RC-52 tumor-bearing mice, either untreated or treated with various activities of ^225^Ac-hG250 or ^177^Lu-hG250 (30 µg). The red line indicates the mean.

**Figure 4 pharmaceuticals-15-00570-f004:**
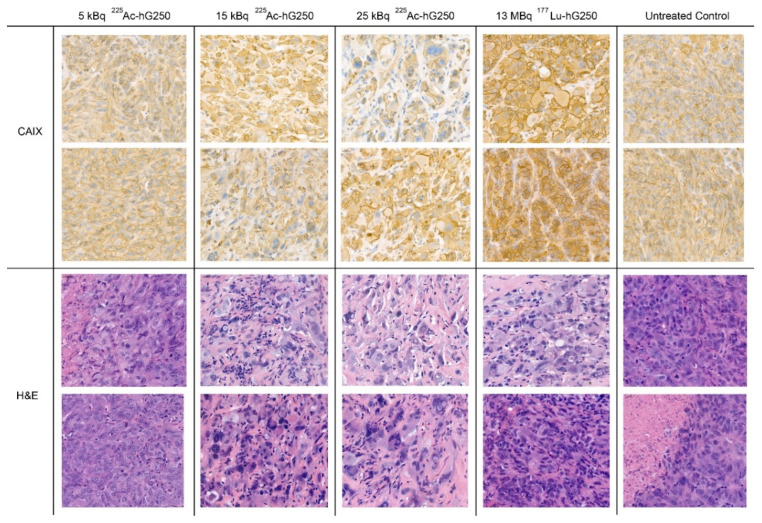
Representative SK-RC-52 tumor slices 125–150 days post-inoculation from mice treated with ^225^Ac-hG250, ^177^Lu-hG250, or untreated control. Upper panel: carbonic anhydrase IX staining, demonstrating a remaining transmembranous CAIX expression after treatment. Lower panel: hematoxylin and eosin staining, displaying radiation damage (enlarged nuclei, atypical nuclei, and intercellular fibrosis) in mice treated with 15, 25 kBq ^225^Ac-hG250 or 13 MBq ^177^Lu-hG250.

**Figure 5 pharmaceuticals-15-00570-f005:**
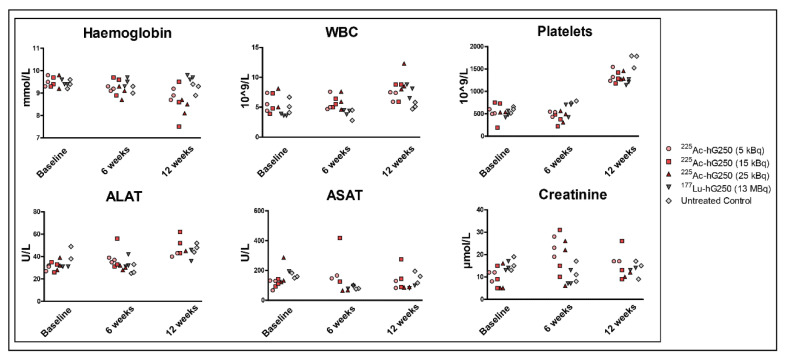
Hematological parameters in non-tumor-bearing mice at baseline, 6 weeks, and 12 weeks after treatment with ^225^Ac-hG250 or ^177^Lu-hG250.

**Figure 6 pharmaceuticals-15-00570-f006:**
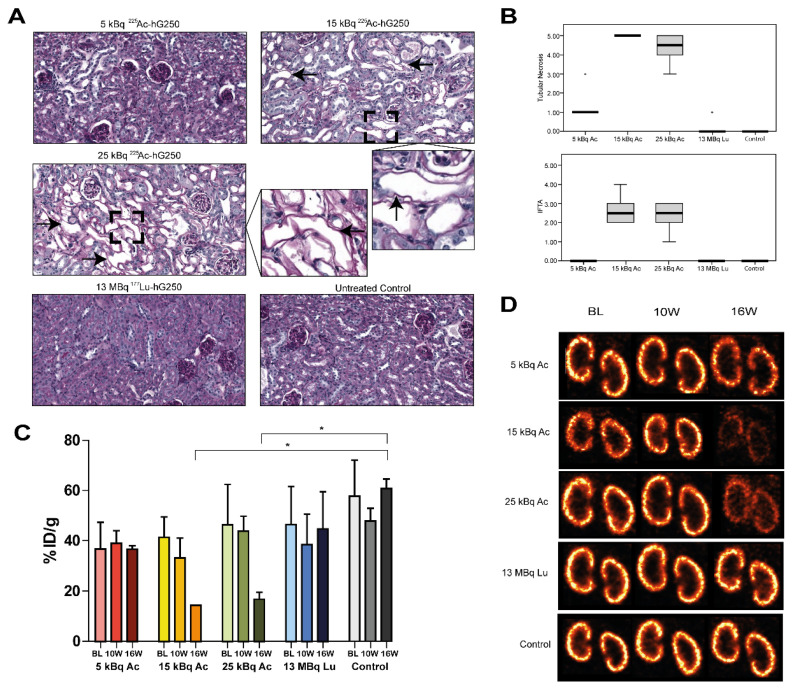
(**A**) Representative kidney slices stained with periodic acid-Schiff from non-tumor bearing mice either untreated or treated with different activity doses of ^225^Ac-hG250 or ^177^Lu-hG250, 18 weeks post-treatment. Arrows indicate tubular denudation. (**B**) Scoring of IFTA and tubular necrosis of the kidneys in treated mice. (**C**) Quantative uptake of ^99m^Tc-DMSA in kidneys of treated mice, expressed as %IA/g. * = *p* < 0.05). (**D**) Visualization of ^99m^Tc-DMSA SPECT imaging in treated mice. IFTA = interstitial fibrosis and tubular atrophy.

## Data Availability

The datasets generated during and/or analyzed during the current study are available from the corresponding author on reasonable request.

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
