# Peer review of "Carbonic Anhydrase IX-Targeted α-Radionuclide Therapy with 225Ac Inhibits Tumor Growth in a Renal Cell Carcinoma Model"

_pharmaceuticals, 2022, doi:10.3390/ph15050570_

Round 1

Reviewer 1 Report

The group of Sandra Heskamp and co-workers has evaluated the therapeutic potential of 225Ac-labeled anti-CAIX antibody hG250 with the 177Lu labeled version in reference. The manuscript covers radiolabeling and in vivo characterization such as ex vivo bio distribution and therapeutic studies. Some histological evaluation was performed to value organ damage and tumor destruction efficiency. The authors found out that biodistribution was similar between the 177Lu and 225Ac antibody, but showed slight differences in tumor/blood ratio and organ accumulation (spleen, liver). A comparable therapeutic efficiency was shown for the animal group treated with 13 MBq of the 177Lu-labeled antibody as well as labeled with 15 and 25 KBq of 225Ac, respectively. There was no treatment effect provable for the group that received 5 KBq of 225Ac antibody or, of course, the untreated control group.

Targeted alpha therapy is of highest interest and relevance for radiopharmaceutical research. The manuscript in general is logically structured, well written and presents results and discussion in a very clear manner. Nevertheless, there may be some minor comments that can be addressed for a minor revision before publication in “pharmaceuticals”.

Comments:

  • Regarding your ex vivo bd setup: in my mind, it would have been beneficial to show one more timepoint to really proof that the maximum uptake is reached at 7d p.i. ; you may comment on that
  • line 102: (18.7 ± .7) – value is missing
  • Why is the uptake of 225Ac-labeled hG250 in spleen/liver higher? Is the molar amount of protein comparable? Did you adjust protein concentrations by adding unlabeled ab? Please clarify – in the experimental section it is written that you used 30 µg each – is this realized by adding more protein than needed to the labeling reaction? Referring to max. spec. activities for both constructs the protein amount is 10-fold higher than actually needed, when I calculated correctly (approx. 3 µg for 50 KBq 225Ac and 0.25 µg for 177Lu). Please comment why you have used 30 µg and not only 3 µg each, for example – this may have negatively influenced your observed bd, as well
  • Comment to supporting info: Please add a small title page (at least names, title and affiliations) to you document

Reviewer 2 Report

General Remarks (to be considered for the whole article):

  • Consensus Nomenclature has to be applied throughout the article (e.g. [225Ac]Ac-hG250, actinium-225, 225Ac-labeled etc). --> see also Coenen et al. Nuclear Medicine and Biology 55 (2017) v–xi.
  • The denomination of the radiolabeled antibody should be kept consistent (with or without DOTA) throughout the article.
  • Replace % ID/g by % IA/g (it is an activity, not a dose that is applied and “dose” is misleading in the field of radiopharmaceutical sciences/nuclear medicine).
  • Use β¯ (beta minus) not just β

Abstract:

  • Line 31: showed weight loss at the end of experiment --> the day of the “end of the study”should be indicated.
  • The amount of injected antibody (substance) should be given.
  • The conclusion of the abstract is not underlined by the results. Why is alpha therapy better if it causes nephrotoxicity?

Introduction:

Bone marrow toxicity is reduced when exchanging 177Lu by 225Ac only when bone marrow metastases are existent. If the bone marrow toxicity is a result of a long circulation time of the radioharmaceutical, why should 225Ac be less toxic than 177Lu? It would destroy blood cells even with more potency. This section has to be revised and the statements underlined with respective citations of adequate literature.

Therefore, it is paramount to 80 find the therapeutic window for targeted α-therapy that results 81 in minimal toxicity [11].

--> if ever it makes sense to use actinium-225 with an antibody that circulates for such a long time. This section should contain more critical view of such combinations.

Methods:

Based on which consideration was the activity chosen. In clinics, patients are for example injected with about 7.4 GBq 177Lu-PSMA-617 and about 8 MBq 225Ac-PSMA-617. This means a factor 1000 in activity. Would it not have been wise to use the lutetium-177 in this study in a 1000-fold higher amount of activity?

Line 385: Do not start a sentence with a number. If not otherwise possible to change the sentence, the number has to be given as a word.

In terms of 225Ac labeling it should be better described, when the TLC was measured to know whether 225Ac (via daughters) or daughters were measured (the former would require a time lag from development to measurement). (as mentioned on line 440).

SK-RC-52 cells: This cell line has to be spcified with including the indication of the provider.

Endpoints: tumor volume ≥ 2 cm3, This is a huge volume given the fact that the Balb/c nude mice are very small. In most countries this would not be allowed. Why was this chosen and how justified?

Line 501: «all mice were killed» - a more adequate word should be used “sacrificed”, “euthanized”.

Results:

“Tumor accumulation was significantly higher in the 225Ac-hG250 group compared to the 177Lu-hG250 group” and also for the blood you observe this à do the authors have an explanation for this finding? This should be discussed in the discussion part as the common believe was so far that the distribution of 177Lu- and 225Ac-labeled compounds would be equal. It appears, however, that 225Ac-labeled compound remains longer in the blood (which probably leads to increased tumor uptake?

Lines 112-115: it does not seem to be at the right place in the section “biodistribution study”? Or if you want to mention it as a finding of the dissected organs, this should be clearly indicated as such.

It would be considerably easier for the reader if the therapy (with tumor-bearing mice) and the tolerability with non-tumor-bearing mice were put in separate paragraphs and also reported separately. It is otherwise difficult to understand what was exactly done with which groups of mice.

Line 146: “Survival was significantly prolonged in mice treated….” – The median survival time or a number of a value that underlines this statement should be given.

Figure 3: A line with the average tumor volume in color would be very helpful and make the graphs nicer.

Line 152: “different activity doses” – replace this term by “various activities” (as it not a dose).

Line 180: “Blood parameters showed no significant differences in hemoglobin, leucocyte, ALAT, ASAT or creatinine in comparison to baseline measurement across all treatment groups. – Here one should distinguish between blood plasma parameters (such as ALAT etc) and blood counts such as leucocytes. To use “Blood parameters” for all doesn’t look right.

Figure 5: A line with the average value should be included in the graphs. Furthermore, the titels and axis description should be written in larger font to make it readable.

Discussion:

The unequal distribution of 177Lu- and 225Ac-labeled antibody should be discussed and a potential explanation given for this discrepancy.

225Ac-based immunotherapy will stay highly problematic in terms of side effects due to the decay of 225Ac and its daughters outside of the tumor cells. This should be discuss in more detail. The disadvantage of using 225Ac with antibodies as compared to their use with small molecules (e.g. PSMA-617) should also be critically discussed.

Reviewer 3 Report

The authors submitted a manuscript describing the pre-clinical evaluation of an hG250 antibody for Carbonic Anhydrase IX and the therapy trials by comparing 117Lu and 225Ac. They compared both radioconjugates and moreover pointed out the behavior of 213Bi as part of the 225Ac-decay chain.

The manuscript is written very well. I liked to read it and I have only some small comments to revise:

Line 29: (p<.05) write: (p < 0.05) this is several times found in the manuscript (lines 98, 99, 149, 215 …)

Line 111: add space character (n = 5)

Line 181: add comma after ASAT

Line 239: “two- to ten-fold” space character is missing

Line 249: add comma after “15”

Check and revise the style of the references, they are not according to the manuscript guidelines.

Round 2

Reviewer 2 Report

The authors have addressed the questions to my satisfaction.